# A Challenging Multimodal Video Summary: Simultaneously Extracting and Generating Keyframe-Caption Pairs from Video

**Keito Kudo**[1] **Haruki Nagasawa**[1] **Jun Suzuki**[1] **Nobuyuki Shimizu**[2]
[1]Tohoku University [2]LY Corporation

{keito.kudo.q4, haruki.nagasawa.s8}@dc.tohoku.ac.jp jun.suzuki@tohoku.ac.jp
nobushim@lycorp.co.jp

## Abstract

This paper proposes a practical multimodal video summarization task setting and a dataset to train and evaluate the task. The target task involves summarizing a given video into a pre-defined number of keyframe-caption pairs and displaying them in a listable format to grasp the video content quickly. This task aims to extract crucial scenes from the video in the form of images (keyframes) and generate corresponding captions explaining each keyframe's situation. This task is useful as a practical application and presents a highly challenging problem worthy of study. Specifically, achieving simultaneous optimization of the keyframe selection performance and caption quality necessitates careful consideration of the mutual dependence on both preceding and subsequent keyframes and captions. To facilitate subsequent research in this field, we also construct a dataset by expanding upon existing datasets and propose an evaluation framework. Furthermore, we develop two baseline systems and report their respective performance. [1]

## 1 Introduction

The popularity of video sharing platforms has increased, which has resulted in a substantial increase in daily video-watching by individuals. As a result, there is increasing interest in practical video summarization systems that can comprehend video content efficiently, and many previous studies have proposed different automatic video summarization methods to address this demand.

Most early attempts only considered video and image data, and these methods were developed in the vision and image processing community (Apostolidis et al., 2021), e.g., keyframe detection (Wolf, 1996; Kulhare et al., 2016; Yan et al., 2018; Khurana and Deshpande, 2023) and video storyboarding (Zhang et al., 2016). However, the recent trend

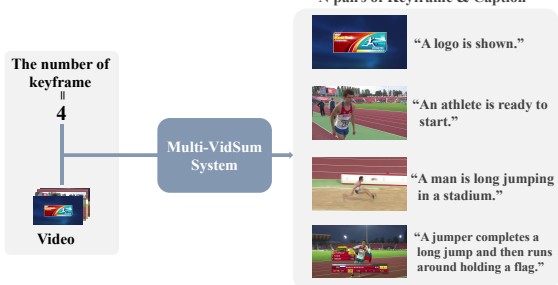

Figure 1: Overview of the Multi-VidSum task. We assume that the number of the output keyframe-caption pairs is given (e.g., 4) depending on summary display space and user preference.

has shifted to multimodal video summarization, which requires joint outputs of both image and text pairs, e.g., video captioning (Sun et al., 2022; Koh et al., 2023), dense video captioning (Krishna et al., 2017; Zhou et al., 2018b; Yang et al., 2023b), and video storytelling (Li et al., 2020).

Multimodal video summarization tasks are a high-demand practical application and a challenging research task. Thus, in this paper, we propose a multimodal video summarization task, which we refer to Multi-VidSum task. This task considers more practical situations and challenging perspectives. The Multi-VidSum task requires us to summarize a given video into a predefined number of keyframe-caption pairs and display them in a listable format to grasp the video content efficiently.

We first formulate the task definition and develop evaluation metrics to assess the task appropriately. Then, we construct training and evaluation datasets to process this task. In addition, we develop two baseline systems using the constructed data, thereby providing an effective benchmark for future research and offering insights into handling the target task based on performance comparisons.

The challenge of this task lies in the simultaneous execution of keyframe selection and caption generation while maintaining sufficient consider-

---

ation of their relative importance. Specifically, to generate appropriate captions, we attempt to select keyframes that are well-aligned with the desired content (and vice versa). Thus, the system must consider these interdependencies and make optimal choices for keyframes and captions, which is challenging. Therefore, the proposed task provides a practical setting that bridges the gap with real-world applications.

Our primary contributions are summarized as follows: (1) We propose the challenging and practical Multi-VidSum task. (2) We generate a dataset for this task to facilitate effective training and evaluation of relevant systems. (3) We develop a new evaluation metric for this task. (4) We develop and analyze two baseline systems, thereby providing a benchmark and insights for future research.

## 2 Related Work

Here, we introduce several related studies. Note that other related works (e.g., multinodal generation and keyframe detection) are listed in Appendix A.

**Multimodal summarization**   Multimodal summarization studies have explored incorporating images in document summarization (Zhu et al., 2018). Similarly, in terms of video summarization tasks, previous studies have proposed methods to generate summary results using both images and text (Fu et al., 2021; He et al., 2023). However, in these methods, the text output by the model is extracted from the input automatic speech recognition results or documents; thus, such methods do not generate summaries by considering visual information explicitly. In our research, our goal is to generate summaries by considering visual information and selecting appropriate frames simultaneously.

**Video storytelling**   Video storytelling (Li et al., 2020), which is a promising video summarization task, has limitations in terms of its dataset, model, and evaluation criteria. For example, the relevant dataset is small and inadequate for training and evaluation. In addition, this method relies on gold data to derive the number of keyframe-caption pairs, which results in unrealistic task settings. Also, keyframe detection was not assessed due to an absence of keyframe annotations. Therefore, in the current study, we constructed a large dataset with corrected keyframe annotations from the open domain to address these limitations.

**Dense video captioning**   In the dense video captioning task, events in video segments are detected and captions are provided. For example, a previous study Krishna et al. (2017) introduced the first dataset for this task, i.e., the ActivityNet Captions, and proposed a corresponding baseline system. We extend this task to the proposed Multi-VidSum task, where keyframes serve as compressed segments. In addition, we create a distinct Multi-VidSum dataset by adding keyframe annotations to the existing ActivityNet Captions. The Multi-VidSum task poses a more significant challenge as it requires a precise selection of $N$ keyframe-caption pairs as representative summaries.

**Visual storytelling**   Similar to the Multi-VidSum task, visual storytelling (Huang et al., 2016; Wang et al., 2018c) has been proposed previously. The VisStory task involves generating story text that is relevant to the input images. Although the concept of the VisStory task is similar to that of the Multi-VidSum task, the Multi-VidSum takes the video data as the input; thus, it is a more challenging setting than the visual storytelling task. In addition, we use this dataset to pretrain our baseline model (see Section 5.3.1 for details).

## 3 Proposed Multi-VidSum task

In this section, we define the proposed Multi-VidSum task. The multimodal Multi-VidSum task attempts to present users with a set of keyframes from the input video with relevant captions.

### 3.1 Summary Length

First, we discuss the summary length, which refers to the number of keyframes and their caption pairs to summarize the video data. Determining the appropriate summary length is heavily dependent on factors, e.g., video length and the number of scenes. However, other considerations, e.g., the space available to display the summary, layout constraints, and user preferences to comprehend the summary, also play significant roles. Thus, it is necessary to incorporate situational information alongside the video to estimate the suitable summary length. However, preparing such situational information is difficult; therefore, in this study, we assume that the appropriate summary length $N$ is always given as the input together with the video data (similar to providing a preferred summary length from a user in

an on-the-fly manner.)[2]

## 3.2 Task Definition

In this section, we define the proposed Multi-VidSum task. Here, let $\mathbf{x} = (x_1, \ldots, x_T)$ be a video, where $T$ is the video length, $\mathbf{y} = (y_1, \ldots, y_N)$ is a series of keyframes selected from video $\mathbf{x}$, and $y_i$ is the $i$-th keyframe when viewed from the beginning of the video. Note that the keyframe is a single frame selected from the video. In addition, $\mathbf{z} = (z_1, \ldots, z_N)$ is a series of explanatory text corresponding to the series of keyframes $\mathbf{y}$. Thus, $z_i$ is the caption corresponding to $y_i$. Then, the proposed Multi-VidSum task can be defined as follows:

$$\mathcal{T} \colon (\mathbf{x}, N) \to (\mathbf{y}, \mathbf{z}). \tag{1}$$

As shown in Figure 1, the system output can also be interpreted as a sequence of keyframe-caption pairs, i.e., $((y_1, z_1), \ldots, (y_N, z_N))$.

## 3.3 Task Difficulty (Challenge)

There are two primary challenges associated with the proposed task. The first challenge is related to the task's inherent complexity, and the second challenge is related to the difficulty associated with evaluating performance. These challenges are discussed in the following subsection.

### 3.3.1 Task complexity

The greatest challenge involved in solving Multi-VidSum is how we select and generate a set of keyframe-caption pairs simultaneously in an efficient and effective manner. Specifically, the procedures used to select keyframes and generate captions are interdependent. Here, the keyframe information is required to generate appropriate captions, and the caption information is indispensable in terms of selecting appropriate keyframes. This implies that determining one without using the other one may induce a suboptimal solution at a high probability. To the best of our knowledge, the interdependence among the different modalities (i.e., image and text) is unique and does not appear in existing tasks. Thus, we must consider a new method to solve this property effectively and efficiently when developing related video summarization methods.

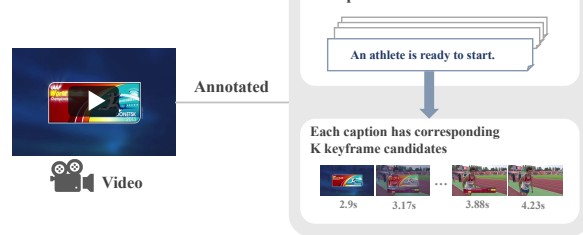

Figure 2: Overview of the Multi-VidSum dataset. Each video comprises $M$ (reference) captions, and each caption has $K$ (reference) keyframe candidates.

### 3.3.2 Evaluation difficulty

The dataset structure for this task (Figure 2) includes videos with $M \geq N$ captions, where each caption has $K$ keyframes. These keyframe-caption pairs are basic building blocks used to construct the video summaries, thereby resulting in multiple valid summarization options for each video. Thus, evaluating the performance of video summarization systems is challenging because the general approach of comparing system outputs to a single reference is feasible. To address these difficulties, we establish an evaluation framework that accounts for these issues (Section 3.4). As a result, we can create an environment that is conducive to assessing the quality and effectiveness of video summarization systems.

## 3.4 Evaluation Criteria

The following are descriptions of the evaluation criteria for each type of output; keyframe and caption.

### 3.4.1 Evaluation criterion for keyframe detection

Here, we assume that each video has $M$ reference captions, with each caption having $K$ keyframe candidates (see Figure 2). Multiple keyframe candidates are used to capture different scenes that align with each caption. We calculate a matching score based on exact matching to evaluate the predicted keyframe list against the reference keyframe list. This involves determining whether the predicted keyframes match the answer keyframes precisely. We define the **aligned keyframe matching (AKM)** score as the maximum matching score over all possible sub-lists of the reference keyframes. However, exact matching (**AKM$_{\text{ex}}$**) can be strict; thus, we also introduce a flexible matching score (**AKM$_{\text{cos}}$**) based on the cosine similarity between the predicted and reference keyframe feature vectors. Note that both

---

[2]Similar discussions and solutions have been made relative to the document summarization task ( (Over and Yen, 2003)).

| | |
|---|---|
| Number of videos | 12,009 |
| Average Number of key-frames per caption | 14.72 |
| Average Number of captions per video | 4.8 |
| Average Number of words per sentence | 13.20 |

Table 1: Statistics of Multi-VidSum dataset

metrics range from 0-1, where higher values indicate better keyframe selection performance.[3]

### 3.4.2 Evaluation criterion for caption generation

The generated captions are evaluated in terms of two distinct metrics, i.e., **METEOR** (Lavie and Agarwal, 2007) and **BLEURT** (Sellam et al., 2020).[4] In this evaluation, the top $N$ keyframes predicted by the model and their corresponding captions based on the highest AKMcos scores are selected. Then, the selected keyframes and captions are used as references to calculate the AKMex, BLEURT, and METEOR values.

## 4 Multi-VidSum Dataset

We constructed a dataset for training and evaluating the Multi-VidSum task in machine learning approaches. To impact the community more, we constructed a dataset for the Multi-VidSum task by expanding ActivityNet Captions (Krishna et al., 2017), which is commonly used, including the shared task of Dense-Captioning Events in Videos.[5] We tasked crowd workers to add keyframe annotations that align with the captions in the ActivityNet Captions.[6] Note that the original ActivityNet Captions dataset has no keyframe annotation; thus, including the additional annotations to handle the proposed Multi-VidSum task is considered a unique contribution to the field.

### 4.1 Statistics

The ActivityNet Captions dataset contains approximately 15,000 videos. We excluded some videos to annotate for several reasons, e.g., unavailable videos. Thus, the resultant annotated video becomes about 12,009. Table 1 shows the statistics of the dataset generated for the Multi-VidSum task.[7]

---

[3]Refer to Appendix B.1 for a precise definition and additional details.

[4]For detailed information about METEOR and BLEURT, please refer to Appendix B.2.

[5]http://activity-net.org/challenges/2019/tasks/anet_captioning.html

[6]Refer to the Ethics Statement for information regarding wages paid for this annotation.

[7]The detailed process is described in Appendix C.

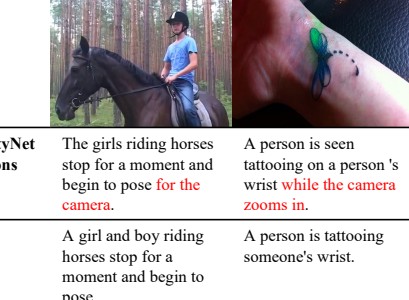

| | | |
|---|---|---|
| **ActivityNet Captions** | The girls riding horses stop for a moment and begin to pose for the camera. | A person is seen tattooing on a person 's wrist while the camera zooms in. |
| **Ours** | A girl and boy riding horses stop for a moment and begin to pose. | A person is tattooing someone's wrist. |

Figure 3: Examples of captions in ActivityNet Captions dataset that mention the cameraman and camera zoom, which are unrelated to the content shown in the given keyframe.

With the crowdsourced keyframe annotations, we obtained an average of 14.72 keyframe annotations per caption.

### 4.2 Reliability Check

We asked a trained annotator (rather than the crowd workers) to assess the matching of the annotated keyframes and the corresponding captions in a random sample of 196 videos. The trained annotator found that 3.87% of the keyframe annotations represented annotation errors. However, we consider that this error rate is sufficient; thus, the generated dataset is sufficiently reliable to train and evaluate systems designed for the Multi-VidSum task.

We also attempted to increase the reliability through automatic annotation filtering. Here, we calculated the image features for all frames using the pretrained model (Zhou et al., 2018b), and then we eliminated annotated keyframes if the distance between the centroid of the image features for all annotated keyframes assigned to a single caption and the image feature of each annotated keyframe was relatively large. For example, we reduced the variance of the image features for the annotated keyframes from 0.0536-0.0166. In this case, the error rate of the remaining annotated keyframes evaluated by the expert annotator was reduced from 3.87%-1.76%.

### 4.3 Test Set Refinement

We constructed our dataset using the caption annotations provided in the original ActivityNet Captions dataset. However, some of the captions in the original dataset are of low quality; thus, there was a concern that an effective and reliable evaluation of the system's performance would not be possible. For example, it refers to the behavior

of the cameraman outside of the frame, e.g., the zoom of the camera. In addition, as discussed in (Otani et al., 2020), this issue is also linked to the limited variety of actions present in ActivityNet Captions. To address these issues, we engaged a reliable data creation company with native English speakers to re-annotate the correct captions for a subset of the original validation set to create a test set with high-quality captions for a random sample of 439 videos. Detailed information about the instructions provided to the annotators is given in Appendix C.2.[8]

A comparison of the captions before and after the re-annotation process is shown in Figure 3. Through the re-annotation process, we qualitatively confirmed a reduction in terms of the references to content not observable in the video, e.g., the cameraman. Thus, we believe that we can perform more accurate evaluations by using this newly annotated test set. For a more detailed analysis of the test set (i.e., action diversity, vocabulary diversity and caption information), please refer to Appendix C.4. Note that the newly annotated dataset was used as the test set in all experiments discussed in this paper.

# 5 Baseline Systems

Here, we propose two baseline systems to tackle the proposed task. To address the challenges discussed in Section 3.3.1, we present two different baseline systems: **Iterative refinement model**, which iteratively selects keyframes and generates captions alternately and **Simul-determination model** that selects and generates all keyframes and captions collectively. In Simul-determination model, given the anticipated difficulty of simultaneous selection and generation of all keyframes and captions, we implement a method that incorporates a pretrained image captioning model. By comparing the outcomes of these two baseline systems, we seek to provide insights to researchers interested in undertaking future investigations of related tasks.

## 5.1 Iterative refinement model

As our first baseline model, we propose Iterative refinement model that divides the task into multiple subtasks and processes them individually and iteratively using different experts. In the following, we provide a comprehensive overview of the Iterative

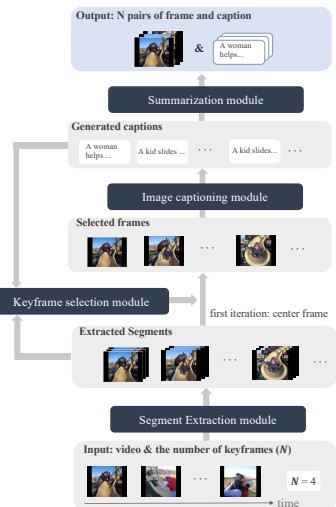

Figure 4: System overview of Iterative refinement model.

refinement model.[9]

### 5.1.1 System descriptions

**Overview** Figure 4 shows an overview of the Iterative refinement model, which comprises four modules, i.e., the segment extraction, image captioning, keyframe selection, and summarization modules. The system takes a video as input and produces $N$ keyframes and captions according to the following process.

1. The segment extraction module divides the video into segments, where a segment refers to a semantically similar and contiguous timespan within the video.
2. The image captioning module generates a caption for the median frame of each segment.
3. The keyframe selection module selects a frame that best matches the generated caption for each segment.
4. The image captioning module then generates another caption for each selected frame. Note that steps 3 and 4 are repeated $l$ times (**iterative refinement**.[10]) The parameter $l$ is fixed at 4 for all settings in this study.
5. Finally, the summarization module selects $N$ keyframe-caption pairs as the system's final output.

In the following, we describe each module in detail.

---

[8]See Ethics Statement about the wages paid for this work.

[9]For detailed experimental configurations and information on reproducing our experiments, refer to Appendix D.

[10]Refer to Appendix E.1 for information regarding the impact of this approach.

**Segment extraction module** The segment extraction module divides the input video into segments. We extract segments first to prevent redundant computations to reduce computational costs because many frames are very similar, especially if the time of their appearance is close. We also use the segment information in the final keyframe-caption pair selection phase.

We use PySceneDetect (Castellano, 2014) and Masked Transformer Model (**MTM**) (Zhou et al., 2018b) to extract segments. PySceneDetect is a video processing tool to detect scene changes, and the MTM includes the functionality to extract segments. Using these tools, we prepare $N$ or more segments by combining the 20 segments extracted by the MTM with those created by PySceneDetect.[11] In addition, the MTM calculates a score that gauges the quality of each segment.[12] This score is then used in the summarization module.

**Image captioning module** To generate captions for the candidate frames, we use pretrained image captioning models, i.e., Fairseq-image-captioning (Krasser, 2020), ClipCap (Mokady et al., 2021), Clip-Reward (Cho et al., 2022), and InstructBLIP (Dai et al., 2023). We also use Vid2Seq (Yang et al., 2023b), which is a dense video captioning model, to generate captions.[13] Note that Vid2Seq is employed not for the direct generation of summaries but only for generating captions for the selected frames. The notable difference between Vid2Seq and the other image captioning models is that Vid2Seq takes the entire video and automatic speech recognition (ASR) result as the input to generate a caption, whereas the image captioning models only take images as the input. Here, Vid2Seq is used to investigate the effectiveness of audio information and the context of the video for the caption generation process. These models also calculate a score that reflects the quality of the generated captions, and this score is used in the subsequent summarization module. Refer to Appendix F for detailed information about these models and scores.

**Keyframe selection module** The keyframe selection module takes a segment and a caption as

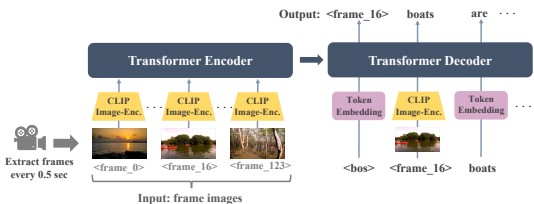

Figure 5: Overview of our Simul-determination model

its input to predict which frames in the segment match the caption. Specifically, this module uses a model that performs binary classification to determine whether each frame in the segment is a keyframe or not. The architecture of this model is based on a bidirectional LSTM. Additional details about this model are given in Appendix D.2.

**Summarization module** The summarization module selects $N$ keyframe-caption pairs from the candidates for each segment. To determine the $N$ keyframe-caption pairs, we consider the sum of the segment score (computed by the segment extraction module) and the caption score (computed by the image captioning module). We apply a dynamic programming technique to select the optimal $N$ keyframe-caption pairs while minimizing time overlap among the segments. The selected keyframes and captions are then used as the final output of the system.

## 5.2 Simul-determination model

The Iterative refinement model generates captions and selects keyframes separately, which means that it does not consider the context of the caption or keyframe. Thus, we also propose a model that generates captions and keyframes simultaneously to address this limitation.

### 5.2.1 Overview

This model is based on the transformer (Vaswani et al., 2017) encoder and decoder architecture with several modifications to make it suitable for the target task. Figure 5 shows an overview of the model. The encoder takes as input the feature vectors of all frames in the video (obtained by down-sampling every 0.5 seconds) as the input, where the image encoder of CLIP (Radford et al., 2021a) is used to create the feature vectors. Then, the decoder produces a sequence of $N$ frame indices and their captions, thereby predicting a keyframe and generating a caption $N$ times. Although the keyframe and caption are different modalities, this model

---

[11]We exclude overly long segments (longer than 75% of the entire video) and reduce the score for segments that overlap with other segments.

[12]As for PySceneDetect scores, they are uniform and fixed at 1.

[13]We used the Vid2Seq model reimplemented in Pytorch (Yang et al., 2023a).

generates and selects them as a single sequence, which allows the model to consider the context while generating captions and selecting keyframes. If the input is a text token, the decoder receives the embedding corresponding to the token, and if it is a frame, the decoder receives the features created by CLIP. We modify the architecture to fit the target task, which is described in the following.

### 5.3 Architecture Changes

**Gate mechanism**   This model follows a repetitive process whereby a keyframe is predicted for each caption, and corresponding text is then generated. This cycle is repeated for a total of $N$ iterations. Thus, the model should predict a keyframe when the input is the <bos> token, but a text token otherwise. To enable this, we introduce a gate mechanism that determines whether to predict a frame index or a text token according to the input token. The formulation of the gate mechanism is given in Appendix G.4.2.

**Pointer mechanism**   In the proposed method, it is necessary to select a keyframe from the input frames in the decoder. However, applying cross-attention only allows the model to use the encoder input information (i.e., frames) indirectly for prediction. Thus, we also apply a pointer mechanism inspired by CopyNet (Gu et al., 2016). Here, when predicting a keyframe, we calculate the cosine similarity between the last hidden state of the decoder and each hidden state of the encoder, and the softmax function is applied to obtain the probability distribution that the frame is the keyframe. The corresponding formulation is given in Appendix G.4.2.

#### 5.3.1 Training and inference strategy

**Loss function**   In Multi-VidSum, accuracy in selecting keyframes and the quality of generated captions are equally important, and we cannot prioritize one over the other. Thus, we optimize the keyframe prediction loss and generated captions loss with equal importance. To achieve this, we calculate the cross-entropy loss independently for both the keyframes and captions, and we minimize their sum. The corresponding formulation is given in Appendix G.4.3

**Pseudo video dataset pretraining**   To improve the caption quality and adapt the model to the video inputs, we perform pseudo video dataset pretraining. We use two image caption datasets, i.e.,

MS COCO (Chen et al., 2015) and Visual Storytelling (Huang et al., 2016)). Each training instance is created according to the following procedure.

1. We select $N$ sets of images and captions from the dataset.[14]
2. We randomly divide the input sequence length of the encoder into $N$ spans.
3. We then select a single index at random from each span. Note that these indices are defined as the indices of the $N$ keyframes.

For each span, the input to the encoder comprises the same image features. However, except for the image feature that corresponds to the index of the keyframe (selected in the above procedure), we apply noise to the features. Additional details about this process are given in Appendix G.3

The model goes through two pretraining phases, where it initially uses a pseudo dataset based on MS COCO and then Visual Storytelling. This two-phase pretraining strategy is employed because Visual Storytelling includes stories that connect each image and caption pair; thus, the content of the dataset is similar to video data.

**Fine-tuning**   After the pretraining phase is complete, we proceed to fine-tune the model using our Multi-VidSum dataset. We sample eight patterns of $N$ keyframes and the corresponding captions from each video with $N$ or more captions. [15]

**Inference**   In the inference process, we first use the pretrained image captioning model described in Section 5.1 to generate captions for all sampled frames. Then, all the sampled frames are input to the model's encoder, while the frame and generated caption pairs are provided as input to the model's decoder to calculate a score based on the likelihood-based. (refer to Appendix G.2 for additional details). Finally, the system determines the final output by identifying the list of $N$ keyframes and captions scored highly by the model. Note that there are numerous combinations of $N$ keyframe-caption pairs; thus, calculating the likelihood for all possible combinations is computationally expensive. In addition, the ideal keyframes and captions to be selected depend on the preceding and subsequent keyframes and captions. To address

---

[14]For the MS COCO dataset, we sample $N$ images and their accompanying captions. For the Visual Storytelling dataset, we use the first $N$ images and captions of a story.

[15]Multiple keyframes are annotated to a single caption (Section 4.)

| | | Keyframe | | Caption | |
|---|---|---|---|---|---|
| | | AKM$_{ex}$ | AKM$_{cos}$ | BLEURT | METEOR |
| Iterative refinement | Fairseq-image-captioning | 38.15 | 80.14 | 31.66 | 9.88 |
| | ClipCap | 38.95 | 79.32 | 30.61 | 10.12 |
| | CLIP-Reward | 38.10 | 80.20 | 37.91 | 10.66 |
| | InstructBLIP (zero-shot) | 37.13 | 78.17 | 35.15 | 12.69 |
| | InstructBLIP (few-shot) | 38.27 | 78.55 | 36.67 | 13.46 |
| | Vid2Seq | 42.35 | 81.20 | **45.16** | 15.17 |
| Simul determination | self | 37.93 | 79.22 | 34.30 | 9.22 |
| | Fairseq-image-captioning | 40.21 | 81.30 | 32.44 | 10.25 |
| | ClipCap | 42.03 | 81.80 | 30.84 | 10.36 |
| | CLIP-Reward | 42.43 | 82.01 | 35.28 | 10.49 |
| | InstructBLIP (zero-shot) | **43.22** | 81.80 | 36.94 | 13.42 |
| | InstructBLIP (few-shot) | 41.86 | **82.04** | 37.35 | 13.75 |
| | Vid2Seq | 42.31 | 81.30 | 44.85 | **15.35** |

Table 2: Performance of baseline models on the test set of the Multi-VidSum dataset. Here, "self" means the performance when the Simul-determination model generates captions itself without using pretrained image captioning model results. "Fairseq-image-captioning", "ClipCap", "CLIP-Reward", "InstructBLIP", and "Vid-Seq" are the type of pretrained image captioning models. For readability purposes, all values are displayed multiplied by 100. See Section 5.1, 5.3.1, for more details.

these issues, we introduce a beam search algorithm, where the calculation of the likelihood for a single keyframe-caption pair is performed in a single step. The details of the beam search algorithm and the corresponding score are given in Appendix G.1.

### 5.3.2 Model Architecture

As an implementation of the Transformer, we adopt the architecture of Flan-T5 architecture (Chung et al., 2022). Additionally, we integrate the pointer and gate mechanism, as described in Sections 5.3. Additional details and the hyperparameters used during training are described in Appendix G.4.

## 6 Experiments and Results

**Settings** As described in Section 3.4, we employed multiple evaluation metrics to assess the keyframes and captions. Specifically, in this experimental evaluation, we used AKMex and AKMcos for the keyframes and the METEOR (Lavie and Agarwal, 2007) and BLEURT (Sellam et al., 2020) for the captions. For the test set, we constructed a dataset consisting of 439 videos. In addition, to ensure accuracy and consistency, we performed a re-captioning process on the videos (Section 4.3). Regarding the training data, the specifics differ among the baseline models discussed in Section 5.

**Results** Table 2 compares the performance of the baseline models. As can be seen, the Simul-determination model tends to have higher keyframe selection ability (AKM) and caption generation

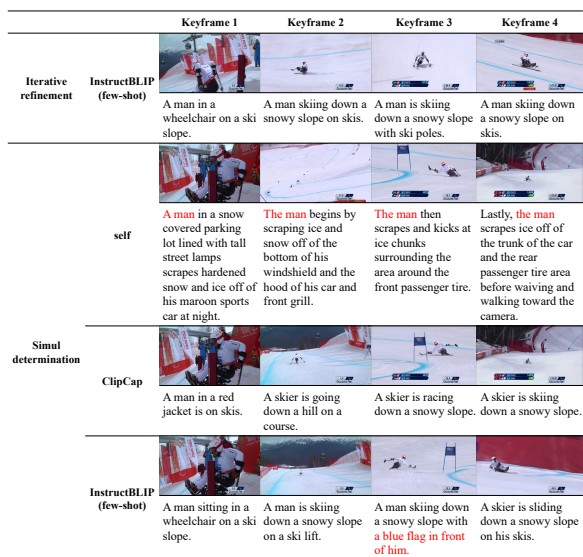

Figure 6: Comparison of summary generated by baseline models for a video.

ability (BLEURT and METEOR) compared to Iterative refinement model. This suggests that Simul-determination model had an advantage over Iterative refinement model in terms of considering the preceding and succeeding keyframes and captions. By comparing the performance of Simul-determination model relative to self-generation versus using captions from a pretrained model, it is clear that both keyframe selection and caption quality realize higher performance when leveraging a pretrained model, which highlights the effectiveness of the proposed approach. Note that the performance of Simul-determination model is heavily dependent on the selection of the pretrained model, and using a superior pretrained model results in higher-quality captions. In addition, when using audio information (i.e., Vid2Seq), the quality of the generated captions was even higher. However, the keyframe selection performance remained relatively consistent, even when the evaluation of generated captions was low. These findings highlight the strengths of Simul-determination model in keyframe selection and caption generation and they underscore the importance of using a pretrained model to realize performance improvements.

## 7 Discussion

### 7.1 Effect of Image Captioning Models

Figure 6 shows the keyframes and captions generated by the Iterative refinement model and Simul-determination model. By comparing the results obtained by the image captioning models, i.e.,

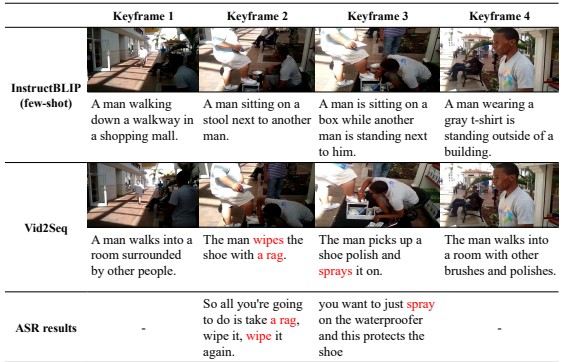

| | Keyframe 1 | Keyframe 2 | Keyframe 3 | Keyframe 4 |
|---|---|---|---|---|
| **InstructBLIP (few-shot)** | A man walking down a walkway in a shopping mall. | A man sitting on a stool next to another man. | A man is sitting on a box while another man is standing next to him. | A man wearing a gray t-shirt is standing outside of a building. |
| **Vid2Seq** | A man walks into a room surrounded by other people. | The man wipes the shoe with a rag. | The man picks up a shoe polish and sprays it on. | The man walks into a room with other brushes and polishes. |
| **ASR results** | - | So all you're going to do is take a rag, wipe it, wipe it again. | you want to just spray on the waterproofer and this protects the shoe | - |

Figure 7: Summaries generated by Iterative refinement model using Vid2Seq and InstructBLIP (few-shot), along with the ASR outputs that appear to influence the captions generated by Vid2Seq.

the ClipCap and InstructBLIP, it can be seen that captions generated by InstructBLIP, which is a high-performance image captioning model, provide more informative summaries by describing finer details within the images effectively. For example, in Figure 6, when comparing the results for Keyframe 3 generated by the Simul-determination model using ClipCap and InstructBLIP, we can observe that the latter caption includes the specific information, e.g., *"a blue flag in front of him."* In addition, the captions generated by using the pretrained models (e.g., InstructBLIP) are more natural and accurate than the self-generated captions. In contrast, the self-generation approach incorporates both definite articles and pronouns for coherence and storytelling, as evident in the use of *"A man"* and *"The man"* in consecutive captions (Figure 6).

### 7.2 Qualitative Differences between Baseline Models

In Iterative refinement model, the keyframes are predicted based on the segments identified by the segment extraction module as described in Section 5.1.1. Thus, we observed a tendency for various keyframes to be selected compared to the Simul-determination model. However, the variation in keyframes does not necessarily correlate with the quality of the summaries. For example, some frames might be less significant, containing only text as titles that are dissimilar to other frames. As a result, the Iterative refinement model exhibits inferior performance compared to the Simul-determination model.

### 7.3 Effect of Audio Information

As shown in Table 2, using Vid2Seq as an image captioning model demonstrated superior performance for the generated captions compared to models that rely solely on image inputs, e.g., InstructBLIP. This improvement can be attributed to Vid2Seq's utilization of audio information during the caption generation process. Figure 7 shows output examples obtained by Iterative refinement model using Vid2Seq and InstructBLIP (few-shot), along with the ASR results that are expected to have influenced the captions generated by Vid2Seq. For example, the video in Figure 7 shows the shoe polishing process. Recognizing the act of shoe polishing requires attention to specific details in the image; thus, it is considered challenging for an image captioning model that solely takes the image as the input to grasp the situation accurately. In contrast, by leveraging audio information, Vid2Seq excels at grasping the nuances of shoe polishing accurately, thereby enabling the generation of captions that incorporate more detailed information. In fact, we found that captions for Keyframes 2 and 3 generated by Vid2Seq include precise details by utilizing specific words extracted from the ASR results taken as the input, e.g., "wipe," "a reg," and "spray".

## 8 Conclusion

In this paper, we have proposed a practical and challenging task setting of a multimodal video summarization, Multi-VidSum. We also extended the ActivityNet Captions with the keyframe annotations obtained by the human annotators (crowd workers). To the best of our knowledge, our dataset is the first large video dataset with keyframe annotation, and its quality was assured by human evaluations. We also proposed evaluation criteria of Multi-VidSum task to appropriately evaluate this task. Extensive experiments were conducted on our dataset using two baseline methods, and we provided a benchmark and relevant insights. We hope our dataset will make new movements in vision-and-language research and accelerate them.

## Limitations

In this work, we proposed a video summarization task based on an actual use case and proposed a corresponding dataset and baseline systems for it. However, our proposed the Simul-determination model does not consider the entire keyframe-caption pairs to generate the optimal summary (despite being somewhat alleviated by the beam search algorithm). To address this issue, it will be necessary to construct a system that generates a summary that considers future information using reinforcement learning.

In addition, as discussed in Section 5.2.1, the proposed Simul-determination model takes every frame downsampled every 0.5 s from the video as input. However, this configuration raises a scalability issue because the number of frames taken as input increases with the length of the video, thereby leading to high computational costs. To address this scalability issue, a potential solution involves integrating a model that identifies salient frames from the video. However, to maintain simplicity in the baseline system, this development is left as a future task in this work.

## Ethics Statement

The initial state of our dataset, which we created to facilitate this study, is the existing dataset, ActivityNet Captions dataset, as mentioned in Section 4. In addition, our models were trained on only the constructed dataset and published datasets that have been used in many previous studies. Thus, our task, dataset, and methods do not further amplify biases and privacy for any other ethical concerns that implicitly exist in these datasets and tasks.

In addition, we used a crowd-sourcing system to annotate the keyframes for the videos in the initial dataset (Section 4). Here, we paid a total of approximately $30,000 USD worth of Japanese yen for this keyframe annotation work. Moreover, as described in Section 4.3, we reannotated the captions in the test data. Here, we additionally paid approximately $10,000 USD worth of Japanese yen for this caption reannotation process, representing $22.8 per video. We believe that we paid a sufficient amount to the workers for this annotation labor.

We disclose some privacy risks and biases in the video because the source of our dataset is the videos submitted to a public video-sharing platform. However, we emphasize that these videos are not included in our dataset as well as the initial dataset of our dataset, and thus, owners of the video immediately delete their videos at any time they want.

## Acknowledgements

We thank four anonymous reviewers who provided valuable feedback. We would like to also appreciate the member of Tohoku NLP Group for their cooperation in conducting this research. We would like to especially thank Kotaro Kitayama, Shun Sato, Tasuku Sato, and Yuki Nakamura for their contributions to the predecessor of this research. This research was conducted as a joint research between LY Research and Tohoku University. Also, part of this research (dataset construction) was conducted with the support of JST Moonshot R&D Grant Number JPMJMS2011 (fundamental research).

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

# A  Other related work

**Multimodal Generation**  Several models for multimodal generation tasks have been proposed based on a causal language model (Sun et al., 2022; Koh et al., 2023). The study by (Sun et al., 2022) is notable for its feature of generating both text and images. Additionally, (Koh et al., 2023) introduced a model that takes text and images as input during language model inference, retrieving images that align with the generated text and producing outputs simultaneously. Our task shares similarities with this study regarding selecting (retrieving) images and generating text simultaneously. However, a notable difference between our task and this task is the visual similarities among candidates of images (frames), where many frames exhibit minimal visual differences in a video. Consequently, considering these subtle distinctions and selecting important frames becomes challenging for our task.

**Keyframe Detection**  Keyframe Detection is a task to select a series of salient frames from videos. This is one of the fundamental and most straightforward video summarizations. The word "salient" here means something conspicuous or representative of the entire video. For example, Yan et al. (2018); Khurana and Deshpande (2023) proposed a method to summarize videos in multiple important images (keyframes). Wolf (1996); Kulhare et al. (2016) proposed a keyframe detection method utilizing optical flow features. Similarly, Fukusato et al. (2016) detected keyframes from anime films. Chen et al. (2018) proposed a video captioning method that can show informative keyframes in video using reinforcement learning.

Narasimhan et al. (2021) proposed a method for query-focused video summarization that summarizes a video into a set of keyframes.

These works take an unsupervised approach to detect keyframes because there are no large datasets available for training Keyframe Detection models in a supervised approach. Despite their approach, we provide a relatively large dataset with keyframe annotations by hand that enables us to train (and evaluate) Keyframe Detection models in a supervised manner.

Displaying selected keyframes on a single page can be an excellent approach for a video summary at a glance. The Multi-VidSum task is a further promising task since it can offer users an easier and quicker understanding of the target video contents by adding captions.

**Video Captioning**  Video Captioning aims to generate a sentence (or a few sentences) describing the overview of a given video. Later, Video Captioning is extended to Dense Video Captioning, which simultaneously detects event locations as video segments and captions for each segment. Note that Dense Video Captioning essentially matches Video Captioning if the videos only consist of one segment (one event) or the videos were split into segments beforehand.

A typical approach to Video Captioning is to use a transformer (Vaswani et al., 2017) that consists of the video encoder and caption decoder (Seo et al., 2022; Li et al., 2022), which can be seen as extensions of image captioning, such as Mokady et al. (2021); Dai et al. (2023). Currently, many studies have proposed techniques to improve the quality of generated captions, e.g., (Yao et al., 2015; Baraldi et al., 2017; Pan et al., 2016; Gan et al., 2017; Pan et al., 2017; Wang et al., 2018b,a; Yan et al., 2020; Bhooshan and K., 2022; Li et al., 2022).

**Video2GIF**  Video2GIF is the task that converts input video into GIF animation. Gygli et al. (2016) proposed the large-scale dataset for the task and show the baseline performance on the dataset. Unlike the Multi-VidSum, Video2GIF is the summarization without using any caption information, so the dataset cannot be diverted to the Multi-VidSum. However, showing the GIF animation as a video summary is an attractive option instead of a keyframe in the Multi-VidSum.

## A.1  Other tasks

Other than the aforementioned tasks, various vision and language tasks such as Visual Question Answering (Antol et al., 2015; Agrawal et al., 2016), Text-Image Retrieval (Kiros et al., 2014, 2018; Lu et al., 2021; Radford et al., 2021b).

# B  Evaulation criteria details

## B.1  AKM Definition

We assume each video has $M$ (reference) captions, and each caption has the $K$ (reference) keyframe candidates, where $M \geq N$ and $K \geq 1$. Figure 2 shows an example. The reason to incorporate multiple keyframe candidates for the reference is that each caption can explain several similar scenes. For evaluating the model of the predicted $N$ keyframe list with the $M$ reference keyframe

set list ($M \geq N$), we calculate the matching score between the two lists. Let $\mathbf{p} = (p_1, \ldots, p_N)$ be a predicted keyframe list by a model. Moreover, $\mathbf{A} = (\mathcal{A}_1, \ldots, \mathcal{A}_M)$ denotes a list of reference keyframe sets. We assume that the sequences in $\mathbf{p}$ and $\mathbf{A}$ are sorted under the chronological order of the given video. Similarly, $\tilde{\mathbf{A}} = (\tilde{\mathcal{A}}_1, \ldots, \tilde{\mathcal{A}}_N)$ is a sub-list of $\mathbf{A}$, whose length is identical to the system output $N$. Note that the meaning of the sub-list here is eliminating $M - N$ reference keyframe sets from $\mathbf{A}$. Then, we introduce a matching function $\mathsf{m}_{\mathsf{ex}}(\cdot, \cdot)$ that receives predicted and reference keyframes and returns a matching score 1 if the inputted two keyframes correctly match and 0 otherwise. Finally, we define **aligned keyframe matching (AKM)** score with exact matching, $\mathsf{AKM}_{\mathsf{ex}}$, as follows:

$$\mathsf{AKM}_{\mathsf{ex}} = \max_{\tilde{\mathbf{A}} \in \mathbf{A}} \left\{ \frac{1}{N} \sum_{i=1}^{N} \max_{a \in \tilde{\mathcal{A}}_i} \left\{ \mathsf{m}_{\mathsf{ex}}(p_i, a) \right\} \right\}, \quad (2)$$

where $\tilde{\mathbf{A}} \in \mathbf{A}$ means taking an $N$ sub-list from $\mathbf{A}$. In Eq. 2, the meaning of computing $\max_{a \in \tilde{\mathcal{A}}_i} \{ \mathsf{m}_{\mathsf{ex}}(p_i, a) \}$ is to check the matching between predicted keyframe $p_i$ and answer keyframe $a$ in the candidate set $\tilde{\mathcal{A}}_i$. We will obtain 1 if $p_i = a$, where $a \in \tilde{\mathcal{A}}_i$ satisfies, and 0 otherwise. The meaning of summation from $i = 1$ to $N$ divided by $N$ is straightforward, just taking the average of inner matching counts over the system output. Note that, in implementation, we can compute Eq. 2 using the dynamic programming technique.

The evaluation by Eq. 2 might be too strict because it is a challenging problem to find exact matching keyframes. To relax the evaluation, we introduce a relaxed matching score $\mathsf{AKM}_{\mathsf{cos}}$ by substituting $\mathsf{m}_{\mathsf{ex}}(\cdot, \cdot)$ as $\mathsf{m}_{\mathsf{cos}}(\cdot, \cdot)$;

$$\mathsf{m}_{\mathsf{cos}}(p, a) = \max \left( 0, \mathsf{Cos}(\bar{\mathbf{v}}_p, \bar{\mathbf{v}}_a) \right), \quad (3)$$

where $\bar{\mathbf{v}}_p = \mathbf{v}_p - \bar{\mathbf{v}}$ and $\bar{\mathbf{v}}_a = \mathbf{v}_a - \bar{\mathbf{v}}$. Moreover, $\mathbf{v}_p$ and $\mathbf{v}_a$ are the image feature vectors for predicted and reference keyframes. $\bar{\mathbf{v}}$ is the mean vector of image feature vectors of a video. $\mathsf{Cos}(\mathbf{x}, \mathbf{y})$ is a function that returns the cosine similarity between given two vectors, $\mathbf{x}$ and $\mathbf{y}$.

## B.2 Caption evaluation criteria

Our evaluation of the generated captions uses two metrics: **METEOR** (Lavie and Agarwal, 2007) and **BLEURT** (Sellam et al., 2020). METEOR stands as a prevalent and extensively employed evaluation metric within the realm of caption generation tasks, and it is precisely the metric we have chosen to adopt in our investigation. BLEURT is a neural language model-based evaluation metric assessing semantic similarity between generated captions and references.

## C   Detail for proposed dataset

### C.1   Data construction procedure

Figure 2 shows the overview of single data in our dataset. Each data consists of three parts; video, a set of captions, and sets of keyframe candidates. Each caption has several keyframes that can match the corresponding caption. The reason for the multiple keyframes for each caption is that the video has the same or very similar frames in one video to consider the multiple candidates. We created the data by assigning caption text to the videos and selecting the keyframes that best match the caption. In more detail, we added the keyframe annotations to the existing captioned movie dataset created by Krishna et al. (2017). We added about 10 keyframe annotations per caption. Specifically, we define the task as selecting the most appropriate time by looking at the video and the caption. The actual working platform used by the cloud worker is shown in Figure 8. On the working platform, the video is placed on top and shows segments by the color bar. The bottom box contains the captions. Each caption has an "add" button to record the keyframe's time. Annotators are required to explore frames (can record multiple candidates) that match the corresponding caption. We set the minimum time unit of the keyframe caption as 0.5 seconds. All cloud workers have to annotate at least one time for each caption.

To extend the dataset, we used Yahoo Crowdsourcing service https://crowdsourcing.yahoo.co.jp/. We carried out the tasks six days a week, which took two months. 352 people joined in total. Each crowd-sourcing task consists of five videos and ten individual workers annotated for each task. On average, about 160 tasks are included in one batch, and three batches are carried out daily.

### C.2   Test set refinement instructions

As detailed in Section 4.3, we conducted a re-annotation of low-quality captions by reliable annotators to ensure a proper evaluation of the task. We asked the annotators to modify the captions so

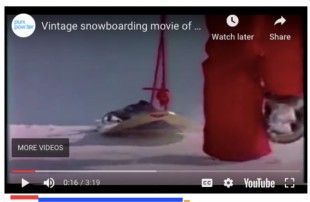

Figure 8: Web application for annotators: annotators can add annotations by pushing buttons.

that they meet the following instructions.

- Each caption should describe the given scene, which consists of a set of images.

- The caption should be a single sentence.

- The caption should be free of spelling and grammar errors.

- The caption should focus on describing the actions of people.

- The caption should contain information that is naturally inferred from the given scene.

- The caption should describe the common elements shared by the majority (more than half) of the 10 or so images extracted for each scene.

### C.3 Train and test split

According to the data split defined in both ActivityNet and ActivityNet Captions, the training data consists of 7,727 samples. We use these 7,727 samples as the training dataset. As described in Section 4.3, the test set comprises 439 videos sampled from the validation dataset, which consists of 4,282 videos defined by ActivityNet.[16]

### C.4 Test Set Analysis

**Action and vocabulary diversity** In video-related tasks, action diversity in the video content is important to realize an effective evaluation (Otani et al., 2020; Yuan et al., 2021). Here, to evaluate the diversity of actions, we investigated the distribution of verbs in the captions (Otani et al., 2020;

---

[16]As the original test set of ActivityNet is not publicly available, we created the test set of our by sampling from the validation set of ActivityNet.

| Rank | 1 | 2 | 3 | 4 | 5 |
|---|---|---|---|---|---|
| Original | **show** 7.9% | see 7.7% | do 3.7% | hold 3.5% | continue 3.5% |
| Refined | play 6.7% | stand 6.2% | hold 5.4% | speak 5.3% | **show** 5.1% |

Table 3: Distribution of the top-5 most frequent verbs in the captions before and after test set refinement. Here, "Original" and "Refined" indicate the distribution of the verbs in the captions before and after refinement, respectively.

| | video | unique verbs | unique nouns | average words per caption |
|---|---|---|---|---|
| Video Storytelling | 15 | 180 | 726 | 11.3 |
| YouCook2 (val) | 457 | 126 | 954 | 8.7 |
| ViTT | 1094 | 1572 | 3057 | 3.1 |
| Ours | 439 | 324 | 1528 | 11.6 |

Table 4: Vocabulary diversity and caption length in existing video summarization datasets (Video Storytelling (Li et al., 2020), YouCook2 (Zhou et al., 2018a), and ViTT (Huang et al., 2020)). These datasets have no keyframe annotations. For the YouCook2 dataset, the test set is not publicly available, and previous work (Yang et al., 2023b) used the validation set as the evaluation data. Thus, we used the validation set for this analysis.

Yuan et al., 2021), and we compared the results before and after re-annotation (Table 3). As a result, we confirmed that the frequency of verbs that do not represent actions (e.g., show) has reduced. Thus, we believe that we can perform more accurate evaluations by using this newly annotated test set.

To confirm the validity of the test set, we also compared vocabulary diversity with existing datasets. Table 4 shows that the newly annotated dataset is inferior in terms of vocabulary diversity compared to ViTT (Huang et al., 2020) because the ViTT is a large-scale dataset and includes a greater number of test samples than ours. In contrast, compared to YouCook2 (Zhou et al., 2018a), which is a dataset of the same scale, our dataset has higher vocabulary diversity. Ideally, it is desirable to increase the number of videos to increase diversity; however, due to budget constraints, we needed to restrict the number of videos used in the test set. Nonetheless, we believe that our dataset contains sufficient diversity in terms of actions and situations for a dataset of this size.

**Caption informativeness** In the context of the video summarization task, the informativeness of a generated summary is a crucial factor (Ermakova

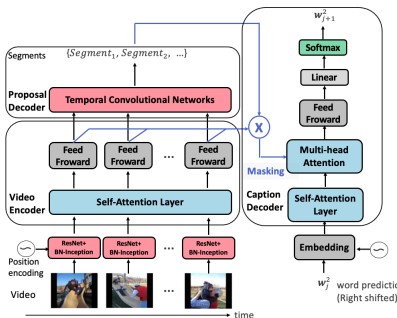

Figure 9: Overview of the MTM model.

et al., 2019) in terms of realizing high-quality summarizations. In this task, the summary is constructed from captions assigned to keyframes; thus, the informativeness of each caption in the test set is crucial in terms of achieving a valid task evaluation. Here, we consider that the informativeness of each caption corresponds to the number of words in each caption (i.e., the length of the caption) because, if the number of words is small, the caption will be insufficient to constitute an effective summary, e.g., omitting information about the subject (i.e., the actor). Thus, we confirmed the validity of the test set by comparing the number of words in the captions between existing datasets. As shown in Table 4, ViTT (Huang et al., 2020), which is a large-scale video summarization dataset, contains many short captions with an average of 3.1 words per caption. In fact, after checking the dataset, we found that it primarily comprises words or phrases, e.g., "Introduction" and "Adding water". In contrast, the captions in our dataset are sufficiently long, with an average of 11.6 words per caption, which is sufficiently informative. Thus, our dataset is considered effective for the proper evaluation of the Multi-VidSum task.

## D   Details of Iterative refinement model

### D.1   Model detail for Segment Extraction Module

As described in Section 5.1, we use Masked Transformer Model (**MTM**) (Zhou et al., 2018b) as the segment extraction module. MTM is a Transformer-based model for the Dense Video Captioning task proposed in (Zhou et al., 2018b). When we train MTM, we follow the instruction written in Zhou et al. (2018b), including the configurations and hyperparameters. Figure 9 shows the overview of the MTM. Here, we only use part of the video encoder and proposal decoder to extract segments.

In Zhou et al. (2018b), they introduced the following four loss function to train the MTM.

- The regression loss $L_r$
- The event classification loss $L_e$
- The binary cross entropy mask prediction loss $L_m^i$
- The captioning model loss $L_c^t$

**The regression loss $L_r$**   The regression loss is the loss function in the proposal decoder to learn the potions of segments:

$$L_r = \text{Smooth}_{l1}(\hat{\theta}_c, \theta_c) + \text{Smooth}_{l1}(\hat{\theta}_l, \theta_l). \quad (4)$$

The MTM model predicts the segments using two variables $\theta_c$ and $\theta_l$. Here, $\theta_c$ refers to the center offset of the segment, and $\theta_l$ refers length offset of the segment in the video. If we decide on these two variables, we can determine a segment. The regression loss is defined as the smooth L1 loss summation between the predictions $(\hat{\theta}_c, \hat{\theta}_l)$ and references $(\theta_c, \theta_l)$.

**The event classification loss $L_e$**   Same as the regression loss, the event classification loss is the loss in the proposal decoder to learn proposal score:

$$L_e = \text{BCE}(\hat{P}_e, P_e). \quad (5)$$

$\hat{P}_e$ and $P_e$ donate the predicted and reference proposal score, and BCE is the binary cross entropy loss. The proposal score is treated as a confidence score for prediction.

**The binary cross entropy mask prediction loss $L_m^i$**   The binary cross entropy mask prediction loss is the loss for connecting the three components, video encoder, proposal decoder, and caption decoder. The loss is defined as:

$$L_m^i = \text{BCE}(B_M(S_p, E_p, i), \\ f_{GM}(S_p, E_p, S_a, E_a, i)), \quad (6)$$

where $S_a$ and $S_p$ indicate the start position of reference and predicted segments and $E_a$ and $E_p$ indicate the end position of reference and predicted segments. $i$ indicates that the frame in the video now model pays attention.

$B_M(S_p, E_p, i)$ is the binary mask function that outputs 1 if and only if the $i$-th frame is included in the reference segments, namely,

$$B_M(S_p, E_p, i) = \begin{cases} 1 & \text{if } i \in [S_a, E_a] \\ 0 & \text{otherwise} \end{cases}. \quad (7)$$

$f_{GM}$ is the smoothed gated mask function between $B_M$ and $f_M$, that is,

$$
\begin{aligned}
f_{GM}&(S_p, E_p, S_a, E_a, i) \\
&= P_e B_M(S_p, E_p, i) \\
&\quad + (1 - P_e) f_M(S_p, E_p, S_a, E_a, i))
\end{aligned} \quad (8)
$$

where $P_e$ is the proposal score.

Next, $f_M$ is the mask function that converts the discrete position information described by Eq. 7 into a continuous one to learn. $f_M$ can be written as:

$$
f_M(S_p, E_p, S_a, E_a, i) = \sigma(g(V_M)), \quad (9)
$$

where $\sigma(\cdot)$ indicates the logit function, and $g$ is the multi-layer perceptron. Moreover, $V_M$ is defined as follows:

$$
\begin{aligned}
V_M = [&\rho(S_p), \rho(E_p), \rho(S_a), \\
&\rho(E_e), B_M(S_p, E_p, i)],
\end{aligned} \quad (10)
$$

where $\rho$ is the positional encoding function, and $[\cdot]$ is the concatenation function.

Finally, the $f_{GM}$ output is passed to the caption decoder. Additionally, thanks to this masking architecture, the caption decoder's loss is passed to both the proposal decoder and the video encoder continuously.

**The captioning model loss $L_c^t$** The captioning model loss is the loss in the caption decoder to learn the caption generation. The loss $L_c^t$ is described as:

$$
L_c^t = \mathrm{BCE}(w_t, p(w_t|X, Y_{\leq t-1}^L)). \quad (11)
$$

$X$ and $Y$ indicate the encoded feature vectors of each frame and caption text, respectively. $w_t$ is the reference caption at the time step $t$. $p(w_t|X, Y_{\leq t-1}^L)$ donates the predicted probability of the $t$-th word of the reference caption.

**Overall loss function** Finally, the total loss function $L$ for MTM is defined as

$$
L = \lambda_1 L_r + \lambda_2 L_e + \lambda_3 \sum_i L_m^i + \lambda_4 \sum_t L_c^t. \quad (12)
$$

### D.1.1 Hyperparameters and Training Configurations

The setting of hyperparameters of the MTM follows the Zhou et al. (2018b). We set the kernel size of the temporal convolution in the Caption Decoder from 1 to 251 and the stride factor to 50. The model dimension of the transformer model

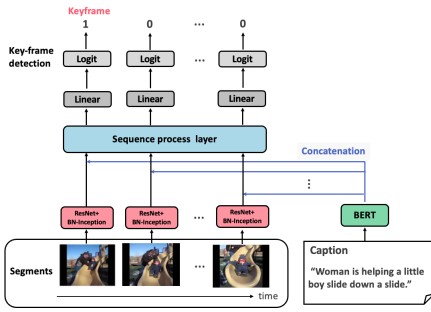

Figure 10: Overview of the Keyframe Detection model.

| Keyframe selection model | |
|---|---|
| Optimizer | Adam (Kingma and Ba, 2015) |
| Learning Rate | 0.0001 |
| Batch Size | 16 |
| Number of Epochs | 20 |

Table 5: The hyperparameters settings of the Keyframe selection Model

is 1024, and the hidden size of the feed-forward layer is 2048. The head size of the transformer is eight, and the number of layers is two. The paraemeter in Eq. 12, we set $\lambda_1 = 10, \lambda_2 = 1.0, \lambda_3 = 1.0, \lambda_4 = 0.25$. We create input image features using both ResNet200 (He et al., 2016) and BN-Inception (Ioffe and Szegedy, 2015). Note that we do not use the validation data for the model selection. We trained the models by the constant iterations and used the models provided by the last iteration. We use the code provided at https://github.com/salesforce/densecap. Also, we used TITAN X (Pascal, Intel (R) Xeon (R)) for training this model.

### D.2 Model detail for Keyframe selection module

Figure 10 shows the overview of the model for the Keyframe selection module (Keyframe selection model). In training for the Keyframe selection model, we use the binary cross-entropy loss as a loss function, namely,

$$
\begin{aligned}
L_{\mathrm{BCE}}(\mathbf{h}) = \sum_{m=1}^{M} -w_n \Big[ &y_m \log \sigma(\mathbf{h}_m) \\
&+ (1 - y_m) \log(1 - \sigma(\mathbf{h}_m)) \Big].
\end{aligned} \quad (13)
$$

Here, $\sigma(x)$ indicates the logit function. Also, $\mathbf{h}_m$ is the output of the linear layer connected after the sequence process layer (Bidirectional LSTM) for

| | Keyframe | | Caption | |
|---|---|---|---|---|
| | AKM$_{ex}$ | AKM$_{cos}$ | BLEURT | METEOR |
| Roop 0 | 33.37 | 78.32 | 36.13 | 13.03 |
| Roop 1 | +4.33 | +0.45 | -0.05 | +0.12 |
| Roop 2 | +4.33 | -0.16 | +0.35 | +0.37 |
| Roop 3 | +4.61 | +0.32 | +0.38 | +0.35 |
| Roop 4 | +4.90 | +0.23 | +0.54 | +0.43 |

Table 6: Ablation study results demonstrate the effectiveness of iterative refinement. The score increment is indicated based on the baseline score from roop 0 (when the center frame of the segment is selected). This result is for the case where InstructBLIP (few-shot) is used as the image captioning model. Iterations roop1 to roop4 correspond to the first to fourth refinements, respectively. For readability purposes, all values are displayed multiplied by 100.

each frame, $y_m$ is the binary reference label that represents whether the frame is the keyframe or not, and $w_m$ is the weight. $M$ is the total number of keyframe captions of the training data. Table 5 shows the model hyperparameter settings of these components. As a pre-processing, the image features from ResNet200 (He et al., 2016) and BN-inception (Ioffe and Szegedy, 2015)) are reduced to 512 dimensions for each by using a linear layer, and then concatenated with text features generated by BERT (Devlin et al., 2019)), whose dimension is 768. Thus, the input dimension becomes 1792 ($512 + 512 + 768$). We used TITAN X (Pascal, Intel (R) Xeon (R)) for training this model.

## E    Ablation study for Iterative refinement model

### E.1    Impact of iterative refinement

Table 6 illustrates the performance change resulting from iterative refinement, where keyframe detection and caption generation are performed alternately. It can be observed that the scores, except for AKMcos, exhibited a tendency to improve with each iteration. However, due to AKMcos being a soft evaluation metric, it is considered that there was no significant change even after the iterations.

### E.2    Impact of keyframe selection module

To assess the effectiveness of the keyframe selection module (Section 5.1.1), we conduct a comparison between the system using the module and a system where the keyframe selection module is omitted, and keyframes is randomly selected. Table 7 shows the results. The experiment results confirmed that the use of the keyframe selection module led to an improvement in performance across

| | Keyframe | | Caption | |
|---|---|---|---|---|
| | AKM$_{ex}$ | AKM$_{cos}$ | BLEURT | METEOR |
| Random | 37.98 | 78.47 | 36.63 | 13.39 |
| Using keyframe selection module | 38.27 | 78.55 | 36.67 | 13.46 |

Table 7: Comparison between the case where the keyframe selection module is used (Using keyframe selection module) and the case where it is not used (Random) in Iterative refinement model. This result is obtained when InstructBLIP (few-shot) is used as the caption generation model. For readability purposes, all values are displayed multiplied by 100.

all evaluation metrics, validating the effectiveness of the keyframe selection module.

## F    Image captioning models detail

### F.1    pretrained image captioning models

We use pretrained image captioning models for both Iterative refinement model and Simul-determination model. Here, we show the details of each model.

**Fairseq-image-captioning**    Fairseq-image-captioning (Krasser, 2020) is a fairseq (Ott et al., 2019) based image captioning library. It converts images into feature vectors using Inception V3 (Szegedy et al., 2016) and generates captions using Transformer. We fine-tuned the model pretrained on MS COCO (Chen et al., 2015) using the keyframe-caption pairs included in our Multi-VidSum dataset.

**ClipCap**    ClipCap (Mokady et al., 2021) is a model that aims to generate high-quality image captions by combining pretrained CLIP (Radford et al., 2021a) and GPT-2 (Radford et al., 2019). ClipCap first converts the input image into a sequence of feature vectors using the visual encoder of CLIP. Then, the sequence of feature vectors is converted by the Mapping Network and used as the prefix of the input to GPT-2 to generate captions. We also fine-tuned this model using the keyframe-caption pairs included in our VideoStory dataset.

**CLIP-Reward**    CLIP-Reward (Cho et al., 2022) is a model that proposes to solve the image captioning task in the framework of reinforcement learning, which enables fine-grained image captioning. The model is trained using the REINFORCE Algorithm (Williams, 1992) with the similarity between the image and the caption calculated by CLIP as the reward, and it can be trained without reference captions. We fine-tuned the image captioning model

trained using CLIP and CLIP as the reward function using the keyframe-caption pairs included in our dataset.

**InstructBLIP** InstructBLIP (Dai et al., 2023) is also an image captioning model that combines pre-trained ViT (Dosovitskiy et al., 2021) and Flan-T5 (Chung et al., 2022) or Vicuna (Chiang et al., 2023) and instruction-tuning was conducted using 26 datasets. In this study, we use the model based on Flan-T5-xxl. Since this model has been reported to perform well on held-out datasets, even in zero-shot, we generated captions with zero-shot and few-shot prompts without fine-tuning. We feed the prompt *"Generate a simple caption for the input image."* to the model for zero-shot and *"Generate a simple caption for the input image like the following examples: <Example: He fights the man in the red suit.> <Example: men are swimming on the lake by the kayak and walking in the lakeside holding kayaks.> Do not copy the examples."* for few-shot (two-shot). The few-shot prompt is composed of randomly selected captions from the training dataset.

**Vid2Seq** Vid2Seq is a pretrained model proposed for the dense video captioning task in Yang et al. (2023b). It takes as input a feature vector for each frame, along with the automatic speech recognition results extracted by Whisper (Dai et al., 2023) and generates tokens that represent the span in the video and captions for that particular span. In this study, we use this model as one of the variants of the image captioning model. Specifically, we pre-generate spans and captions for all videos. During the inference of Iterative refinement model and Simul-determination model, we select a span containing the position (time) of the required frame and use the caption associated with that span. However, the span generated by Vid2Seq does not cover the entire video and there are overlapping spans. Hence, when choosing a caption corresponding to a frame at a particular time, the following algorithm is applied:

1. If there is only one span at that time, use the caption for that span.

2. If there are overlapping spans at that time, prioritize the shorter span (as shorter spans are presumed to contain more localized information).

3. If there is no span at that time, use the caption for the nearest span in time.

In this paper, we use the PyTorch re-implementation of the Vid2Seq code provided by Yang et al. (2023a) and conduct fine-tuning with ActivityNet Captions.

### F.2 Caption score for Iterative refinement model

In the Summarization module of Iterative refinement model, a score reflecting the validity or quality of the captions is used to select a list of $N$ keyframe-caption pairs. This score is primarily based on the likelihood of the captions generated by each model, excluding CLIP-Reward. The CLIP-Reward, on the other hand, uses the similarity between the image and the caption calculated by CLIP (Radford et al., 2021a).

## G Simul-determination model details

### G.1 Beam Search Algorithm

As described in Section 5.3.1, we introduce the beam search algorithm to global optimization. In this process, we repeat the beam search process $N$ times, retaining only the top $W$ (beam size) results at each step and moving on to the next step. For each step, the model calculates the score based on likelihood (see Appendix G.2 for more details about this score) for lists of less than or equal to $N$ keyframes and captions pairs. Finally, among the $W$ remaining results, we select the $N$ pairs with the score based on the highest likelihood as the final output of the model. Algorithm 1 presents the pseudo-code for the beam search algorithm used in Simul-determination model. In the experiment, a beam width of 8 is employed for the Simul-determination model.

### G.2 Score for inference process

To score the selected keyframes and generated captions, we used a score based on the likelihood calculated by Simul-determination model. Specifically, we independently calculated the likelihood of the model selecting the keyframe and the likelihood of the generated caption and used the sum of the normalized values as the score. This is because we consider that the performance of both keyframe selection and caption generation is equally important in this task. The score for $N$ candidates for keyframe and caption is formulated as follows.

$$\text{score}(\mathbf{y}_{\text{cand}}, \mathbf{z}_{\text{cand}}) =$$
$$\frac{1}{N} \sum_{i=1}^{N} \Bigg\{ \alpha \text{norm}_{\text{frame}}(f(y_i | y_{\leq i-1}, z_{\leq i-1}))$$
$$+ (1 - \alpha)\text{norm}_{\text{caption}}(f(z_i | y_{\leq i}, z_{\leq i-1})) \Bigg\}, \tag{14}$$

where $\mathbf{y}_{\text{cand}}$ is the candidate keyframes, $\mathbf{z}_{\text{cand}}$ is the candidate captions $f$ is the model we trained to calculate the likelihood. $\alpha$ is a hyperparameter that controls the balance between the score of the keyframe and caption. We specified $\alpha$ as 0.5 in this study because we considered that keyframe selection and caption generation performance are equally important in this task. $\text{norm}_{frame}$ and $\text{norm}_{caption}$ are the min-max normalization for all frames and corresponding captions likelihoods, respectively.

### G.3 Definition of noise for pseudo Multi-VidSumdataset pretraining

The formulation of the noise for pseudo Multi-VidSumdataset pretraining is as follows:

$$\text{Noise} = \bar{v}\beta x, \tag{15}$$

where $\bar{v}$ is a scalar value obtained by averaging all the values of the image feature in the mini-batch. $\beta$ is a parameter that controls the magnitude of the noise, and in this experiment, $\beta$ was unified to 0.05. $x$ is sampled from a normal distribution $\mathcal{N}(0, 1)$ for each frame.

### G.4 Model and Training Configurations

### G.4.1 Hyperparameters and Training Configurations

Table 8 presents the hyperparameters we use during training. For the parts where the architecture remains unchanged, the initial parameters are taken from the pretrained Flan-T5.[17] This is expected to transfer the language generation ability acquired through pretraining on large amounts of text to solve this task.

Also, as mentioned in Section 5.3.1, we create the pseudo Multi-VidSum dataset and the fine-tuning dataset through random sampling. Consequently, we use different random seeds to construct

---

**Algorithm 1** Beam search algorithm for Simul-determination model

**Input** $NumKeyFrame \in \mathbb{N}$ : Number of key-frame
**Input** $BeamWidth \in \mathbb{N}$ : Beam width
**Input** $Video$ : List of frames in a video

1: $Captions \leftarrow []$
2: **for each** $frame \in Videos$ **do**
3:     Captions.append(ImageCaptioningModel(frame))
4: **end for**
5: $Candidates \leftarrow []$
6: $Beams \leftarrow [[], []]$
7: **for** $i = 1$ to $NumKeyFrame$ **do**
8:     **for each** $Beam_{\text{frames}}, Beam_{\text{captions}} \in Beams$ **do**
9:         $LastTime \leftarrow Beam_{\text{frames}}[-1].time$
10:         **for each** $frame, caption \in Videos, Captions$ **do**
11:             **if** $frame.time <= LastTime$ **then**
12:                 continue
13:             **end if**
14:             $Input_{\text{frames}} \leftarrow$
15:                 $Beam_{\text{frames}} + [frame]$
16:             $Input_{\text{captions}} \leftarrow$
17:                 $Beam_{\text{captions}} + [caption]$
18:             $score = Transformer($
19:                 $Input_{\text{frames}}, Input_{\text{captions}}$
20:             $)$
21:             $Candidates.append($
22:                 $(score, Input_{\text{frames}}, Input_{\text{captions}})$
23:             $)$
24:         **end for**
25:     **end for**
26:     $Beams \leftarrow$
27:         $ScoreTopK(Candidates, BeamWidth)$
28: **end for**
29: **return** Beams[0]

---

[17]We use the published available model on Huggingface Transformers, https://huggingface.co/google/Flan-T5-base

| Pseudo Pre-training (MS COCO) | |
|---|---|
| Number of Training Data | 100,000 |
| Number of Validation Data | 1,000 |
| Optimizer | AdamW |
| | $\beta_1 = 0.9$ |
| | $\beta_2 = 0.999$ |
| | $\epsilon = 1 \times 10^{-8}$ |
| Learning Rate Schedule | Cosine decay |
| Warmup Steps | 1000 |
| Max Learning Rate | 0.00001 |
| Dropout | 0.1 |
| Batch Size | 192 |
| Number of Epochs | 30 |
| **Pseudo Pre-training (Visual Storytelling)** | |
| Number of Training Data | 39,553 |
| Number of Validation Data | 4931 |
| Optimizer | AdamW |
| | $\beta_1 = 0.9$ |
| | $\beta_2 = 0.999$ |
| | $\epsilon = 1 \times 10^{-8}$ |
| Learning Rate Schedule | Cosine decay |
| Warmup Steps | 1000 |
| Max Learning Rate | 0.00001 |
| Dropout | 0.1 |
| Batch Size | 192 |
| Number of Epochs | 30 |
| **Fine-tuning** | |
| Number of Training Data | 23,216 |
| Number of Validation Data | 2902 |
| Optimizer | AdamW |
| | $\beta_1 = 0.9$ |
| | $\beta_2 = 0.999$ |
| | $\epsilon = 1 \times 10^{-8}$ |
| Learning Rate Schedule | Cosine decay |
| Warmup Steps | 500 |
| Max Learning Rate | 0.001 |
| Dropout | 0.1 |
| Batch Size | 192 |
| Number of Epochs | 200 |

Table 8: Hyperparameters and training configurations of Simul-determination model. AdamW is a optiomizer proposed in Loshchilov and Hutter (2019)

the validation set. For the pseudo Multi-VidSum dataset constructed from the Visual Storytelling dataset, we employ the validation set images and captions defined in (Huang et al., 2016). During training, we select the model with the lowest loss on the validation set and proceed to the next training phase. Similarly, during evaluation, we chose the model with the lowest loss on the validation set obtained during fine-tuning. We conduct training of the models using NVIDIA A6000 (48GB memory) and A100 (80GB memory).

### G.4.2 Model Architecture details

**Gate mechanism** The formulation of the gate mechanism is as follows:

$$
\begin{aligned}
p(w_{t+1}|w_{\leq t}) &= \text{Gate}(w_t)p_{\text{frame}}(w_{t+1}|x_{\leq t}) \\
&\quad + (1 - \text{Gate}(w_t))p_{\text{text}}(w_{t+1}|w_{\leq t}) \\
\text{Gate}(w_t) &= \begin{cases} 1 & \text{if } w_t = \texttt{<bos>} \\ 0 & \text{otherwise} \end{cases}
\end{aligned}
$$
$$\tag{16}$$

where $w_t$ is the input token (frame or text token) at time step $t$, $p_{\texttt{frame}}$ and $p_{\texttt{text}}$ are the probabilities of predicting a frame index and a text token, respectively, and $\text{Gate}(w_t)$ is the gate function that controls the prediction modalities.

**Pointer mechanism** The formulation of the pointer mechanism is as follows:

$$
\begin{aligned}
&P_{\text{frame}}(t) \\
&= \text{softmax}(\text{Cos}(h_{dec_t}W_{dec}, H_{enc}W_{enc})),
\end{aligned}
\tag{17}
$$

where $P_{\text{frame}}(t)$ is the probability distribution of the keyframe at time step $t$, $h_{dec_t}$ is the last hidden state of the decoder at time step $t$, $H_{enc}$ is the all last hidden states of the encoder, and $W_{dec}, W_{enc}$ are the weight matrices.

**Other changes** In addition to integrating the pointer mechanism and gate mechanism, several minor architectural changes are made. Specifically, We set the maximum input sequence length of the model to 2048 to handle long videos.[18] Also, to convert the dimension of the input image feature, we include an additional linear layer. Moreover, as indicated in Eq. 17, when predicting the keyframe, the model computes the cosine similarity between the hidden states of the encoder and the decoder. During this process, the hidden state undergoes L2 normalization. Similarly, when predicting the text token, the hidden state is also subject to L2 normalization. Specifically, this is formulated as follows:

$$
P_{\text{token}}(t) = \text{softmax}(\text{norm}(h_t W_{\text{vocab}}^\top)), \quad (18)
$$

where $P_{\text{token}}(t)$ is the probability distribution of text tokens at time step $t$, $h_t$ is the hidden state of the decoder at time $t$, and $W_{\text{vocab}}$ is the weight matrix at the decoder head.

---

[18] The T5 model architecture does not have a limit on the input sequence length, but it is necessary to set a limit for the pointer mechanism and keyframe prediction.

| | Keyframe | | Caption | |
|---|---|---|---|---|
| | AKM$_{ex}$ | AKM$_{cos}$ | BLEURT | METEOR |
| vanilla. | 37.59 | 79.40 | 37.74 | **14.14** |
| vanilla + loss function | 36.79 | 78.96 | **38.00** | 13.97 |
| vanilla + pointer mechanism | **39.24** | 79.34 | 36.88 | 13.66 |
| vanilla + gate mechanism | 36.73 | **79.56** | 37.69 | 14.10 |

Table 9: Results of the ablation study for Simul-determination model. "vanilla" refers to the model trained using the architecture of Flan-T5 (Chung et al., 2022), which is the base architecture of Simul-determination model, with a cross-entropy loss. + loss function indicates the performance when the loss function is changed as described in Section G.4.3. + pointer mechanism and + gate mechanism indicates the performance when the model's architecture is changed as described in Section G.4.2. This result is for the case where InstructBLIP (few-shot) is used as the image captioning model. For readability purposes, all values are displayed multiplied by 100.

### G.4.3 Loss function

The loss function, denoted as $\mathcal{L}$, can be formulated as follows:

$$
\begin{aligned}
\mathcal{L}(\theta, \mathbf{x}, \mathbf{y}_{tgt}, \mathbf{z}_{tgt}) &= \alpha \mathrm{CL}_{frame}(\theta, \mathbf{x}, \mathbf{y}_{tgt}) \\
&+ (1 - \alpha) \mathrm{CL}_{cap}(\theta, \mathbf{x}, \mathbf{z}_{tgt}),
\end{aligned}
\tag{19}
$$

where $\mathrm{CL}$ is a cross-entropy-loss, $\theta$ is the model parameters, $x$ is a input video, $\mathbf{y}_{tgt}$ is the target keyframe, and $\mathbf{z}_{tgt}$ is the target caption. $\alpha$ is a hyperparameter that balances the keyframe and caption importance. We specified $\alpha$ as 0.5 in this study because we considered that keyframe selection and caption generation performance are equally important in this task.

### G.4.4 Ablaition study for architecture and loss function changes

We conducted an ablation study to examine how the changes in the architecture and loss function of Simul-determination modelaffect the performance. Table 9 shows the results of the ablation study. Note that these models are trained only on the Multi-VidSumdataset and not on the pseudo video dataset. As a result of the experiment, the change in the loss function described in Section G.4.3 improved the performance by 0.3 in BLEURT, the introduction of the pointer mechanism described in Section G.4.2 improved the performance by 1.7 in AKM$_{es}$, and the introduction of the gate mechanism improved the performance by 0.16 in AKM$_{ex}$. However, there was no architecture change that resulted in a significant improvement in performance across all metrics. On the other hand, since there was no case where the performance was extremely degraded, we adopted all the changes.