# OpenReview forum: "A Challenging Multimodal Video Summary: Simultaneously Extracting and Generating Keyframe-Caption Pairs from Video"
_EMNLP/2023/Conference — EMNLP 2023 Main_

### Official Review · Reviewer_nTTm · 2023-08-03

**Soundness:** 5

**Excitement:**

4: Strong: This paper deepens the understanding of some phenomenon or lowers the barriers to an existing research direction.

**Paper Topic And Main Contributions:**

This paper proposes a multimodal video summarization task and dataset. They extended Activitynet captions to allow existing models to perform multi-modal video summarization. The authors preserve the dataset's quality and show the proposed task's evaluation process with baseline methods.

**Questions For The Authors:**

- Selected keyframes may be subjective to annotators.
- Can authors check the performance with previous dense captioning models?

**Reasons To Accept:**

- The authors introduced a multimodal video summarization task and dataset contributing to multimodal research.
- The manuscript released a new benchmark dataset for the task of Multi-VidSum with carefully designed refinement and refinement processes. Furthermore, the evaluation criterion for Multi-VidSum is straightforward and simple.

**Reasons To Reject:**

For now, it doesn't seem clear that a multimodal video summarization is more difficult and novel than dense video captioning. The authors should explain the necessity and importance of their proposed task and datasets.

**Reproducibility:**

4: Could mostly reproduce the results, but there may be some variation because of sample variance or minor variations in their interpretation of the protocol or method.

**Reviewer Confidence:**

4: Quite sure. I tried to check the important points carefully. It's unlikely, though conceivable, that I missed something that should affect my ratings.

---

> ### Author Rebuttal · Authors · 2023-08-29
>
> Thank you for your thoughtful review.
>
> ## Importance and difficulty of this task
> **Difficulty:**
> The primary difference between the Multi-VidSum task and the Dense Video Captioning task is that the main objective of the Multi-VidSum task includes summarization.
> In the Dense Video Captioning task, the model is expected to generate captions for **every** event within the video.
> In contrast, the Multi-VidSum task requires **selecting** scenes that are important for comprehending the content of the video.
> Therefore, the Multi-VidSum task presents a **different type of difficulty** from the Dense Video Captioning task.
> A similar relation exists between text summarization tasks and event extraction tasks developed in the NLP field.
> We believe the reviewer will agree that these tasks (text summarization task and event extraction task from text) have essential differences and difficulties, and we hope this statement dispels the concern of the equivalent difficulty between our task (Multi-VidSum task) and Dense Vide Captioning task.
>
>
> As the Multi-VidSum task mainly focuses on summarization, we also mention the difference between this task and existing video summarization tasks.
> In this task, the systems are required to perform summarization through two modalities, which sets it apart from the other video summarization tasks.
> This is a more challenging task setting than unimodal summarization tasks such as[1, 2].
> Specifically, to generate appropriate captions, selecting keyframes well-aligned with the desired content (and vice versa) (l.67-69).
> Consequently, the system must consider these interdependencies and make optimal choices for keyframes and captions, making it challenging (l.69-72).
>
>
> **Necessity, importance, and novelty:**
> The necessity, importance, and novelty of the Multi-VidSum task stem from its orientation towards **more practical** scenarios compared to the Dense Video Captioning task.
> Fundamentally, the objective of the video summarization task is to enable users to comprehend the video's content within a short time.
> Therefore, video summarization is foreseen to have practical utility within video streaming services, and these summaries are expected to be consumed on devices with constrained display areas like smartphones and tablets (l.144-147).
> Given the screen size constraints on these devices, displaying comprehensive video descriptions as required in the Dense Video Captioning task is redundant.
> Similar discussions and solutions have been made in the document summarization task[1] (footnote 2).
>
> Also, the current video summarization tasks[2, 3] often deal with outputs of a single modality.
> This is expected to have low informativeness, making it unsuitable for practical applications.
> Indeed, previous research has reported that multimodal summarization yields higher user satisfaction with the summaries compared to unimodal summarization[4].
> Therefore, in the Multi-VidSum task, the resulting summary will be more easily understandable for users by providing both keyframes and captions as outputs.
> Other video-related tasks[5,6] also require multimodal output.
> However, these tasks are considered insufficient for practical scenarios since the sentences in the summary are created by retrieval-based methods.
> On the other hand, the input of the Multi-VidSum task is only the video, and the captions of the output are generated by the model, making it a more practical task setting.
>
>
>
>
> ## Effect of subjectivity on annotation
> As mentioned in section 4.2, we conducted a reliability check as a post-evaluation of the annotation results.
> In the reliability check, a trusted **third-party** annotator (not crowd workers) evaluated whether each annotated keyframe corresponds to the caption (l.266-269).
> As a result, we found that only $1.76$% of the captions were inappropriate (l.286), indicating that the **impact of subjectivity on the keyframe annotation is limited**.
>
>
>
> ## Performance check with existing dense captioning models
> We greatly appreciate your suggestion.
> Unfortunately, we cannot simply apply existing dense captioning models to our task, so the performance of existing dense captioning models is unavailable.
> This limitation is because the current dense captioning models[7, 8] are designed to generate captions for all events in a video comprehensively.
> They **lack functionalities** to **select $N$ important scenes (keyframes)** and to **extract keyframes**.
>
> ## Reproducibility
> Upon acceptance, we will **provide our data and code as stated in footnote 1**.
> This allows all researchers to validate our paper's experiments using our code.
> We believe many papers guarantee the reproducibility viewpoint in the same manner (by releasing the corresponding source code and data).
> Therefore, our paper should be equal to others in the reproducibility viewpoint and not extraordinarily low.
> We kindly request that reviewers reconsider this point and grade our papers appropriately.
>
>
>
> [1] P. Over and J. Yen. 2003. An Introduction to DUC2003: Intrinsic Evaluation of Generic News Text Summarization Systems.
>
> [2] Evlampios Apostolidis, Georgios Balaouras, Vasileios Mezaris, and Ioannis Patras. 2021. Combining Global and Local Attention with Positional Encoding for Video Summarization. In  IEEE International Symposium on Multimedia (ISM), pages 226–234.
>
> [3] Pinelopi Papalampidi and Mirella Lapata. 2023. Hierarchical3D Adapters for Long Video-to-text Summarization. In Findings of the Association for Computational Linguistics (EACL), pages 1297–1320.
>
> [4] Junnan Zhu, Haoran Li, Tianshang Liu, Yu Zhou, Jiajun Zhang, and Chengqing Zong. 2018. MSMO: Multimodal Summarization with Multimodal Output. In Proceedings of the 2018 Conference on Empirical Methods in Natural Language Processing (EMNLP), pages 4154–4164.
>
> [5] Bo He, Jun Wang, Jielin Qiu, Trung Bui, Abhinav Shri- vastava, and Zhaowen Wang. 2023. Align and Attend: Multimodal Summarization With Dual Contrastive Losses. In Proceedings of the IEEE/CVF Conference on Computer Vision and Pattern Recognition (CVPR), pages 14867–14878.
>
> [6] Junnan Li, Yongkang Wong, Qi Zhao, and Mohan S.Kankanhalli. 2020. Video storytelling: Textual summaries for events. IEEE Transactions on Multimedia, 22(2), pages 554–565.
>
> [7] Ranjay Krishna, Kenji Hata, Frederic Ren, Li Fei-Fei, and Juan Carlos Niebles. 2017. Dense-Captioning Events in Videos. In IEEE International Conference on Computer Vision (ICCV), pages 706–715.
>
> [8] Antoine Yang, Arsha Nagrani, Paul Hongsuck Seo, Antoine Miech, Jordi Pont-Tuset, Ivan Laptev, Josef Sivic, and Cordelia Schmid. 2023. Vid2Seq: Large-Scale Pretraining of a Visual Language Model for Dense Video Captioning. In Proceedings of the IEEE/CVF Conference on Computer Vision and Pattern Recognition (CVPR), pages 10714–10726.

---

### Official Review · Reviewer_wL5Q · 2023-08-05

**Soundness:** 3

**Excitement:**

3: Ambivalent: It has merits (e.g., it reports state-of-the-art results, the idea is nice), but there are key weaknesses (e.g., it describes incremental work), and it can significantly benefit from another round of revision. However, I won't object to accepting it if my co-reviewers champion it.

**Paper Topic And Main Contributions:**

The authors propose a framework for selecting relevant frames and corresponding caption generation from videos that can generate a summary of the video. Two main challenges the author considers for this task are the method of selecting the frame from video segments(iterative and simultaneous approach), caption pairs, and evaluation of the task (aligned keyframe matching score for frames and  METEOR for caption). For this task, the authors also contribute to a new dataset.



**Reasons To Accept:**

1. paper is well-written and chronologically significant.
2. The author collected the dataset for further research in the community.
3. Idea of the Simul-determination model is novel due to its simultaneous generation and performance.

**Reasons To Reject:**

1. For iterative refinement, there is no evidence why only median frame caption works better for selecting keyframes. probably, a better way
would be sampling based on weights (i.e. importance sampling)
2. section 5.2.1, feeding all frames of a video to the encoder may cause an issue in scaling up the model. it would be beneficial if there is a single model to select frames from the video for generating salient keyframes which can be used in both baselines.
3. ablation study for the architectural changes is missing.

**Reproducibility:**

4: Could mostly reproduce the results, but there may be some variation because of sample variance or minor variations in their interpretation of the protocol or method.

**Reviewer Confidence:**

4: Quite sure. I tried to check the important points carefully. It's unlikely, though conceivable, that I missed something that should affect my ratings.

---

> ### Author Rebuttal · Authors · 2023-08-29
>
> We appreciate your constructive comments.
>
> ## Validity of keyframe selection method
> Thank you for your feedback.
> In the iterative refinement model (sec 5.1), **the median frame is used only in the initial step** of the iteration (l.342-343).
> In the following steps, the keyframe selection module predicts a keyframe for generating captions (l.344-346).
> The keyframe selection module, an LSTM-based model, takes a segment (a portion of a video) and a corresponding caption pair as inputs and predicts one single keyframe (l.386-393).
> Through an iterative process of predicting keyframes by the keyframe selection module and re-captioning by the image caption module (l.375-385), we expect a convergence toward more appropriate selections of keyframes.
> Indeed, through the four cycles of this iterative refinement process, we observed enhancements of $+0.54$ points in the caption score (BLEURT) and $+4.90$ points in the keyframe score ($\texttt{AKM}_{\texttt{ex}}$). See Table 5 in Appendix F for more details.
> We hope our answer dispels your concern about not validly selecting keyframes in the iterative method (our baseline system).
>
>
> ## Is it difficult to scale up the model?
> We appreciate your insightful feedback.
> Recent models developed for addressing video-related tasks require the use of all frames from the video as inputs[1, 2, 3].
> Following these previous studies, we have adopted the format of taking all frames as inputs to maintain the simplicity of our proposed model as a baseline system.
> Therefore, as you pointed out, we acknowledge the potential emergence of scalability issues.
> (Note that this challenge is not unique to the model for our task but rather a common concern in video-related tasks.)
> However, the proposed Simul-determination model (sec. 5.2) can work in the same algorithm without taking all frames.
> Hence, the scalability issue could be mitigated by incorporating a model that identifies salient frames from the video, as you suggested.
> Alternatively, we can use an existing method that selects distinct frames[4] to alleviate this issue.
> We hope the reviewer will accept that we keep this scalability issue as a future issue, as it is beyond the difficulty level to solve as the first attempt at our proposed video summarization task since it is not a specific issue of our task, but a common issue for all video-related tasks.
> In the camera-ready version, we will discuss this scalability issue in the limitation section.
>
> ## Missing ablation study
> We appreciate your perceptive comment. In a preliminary experiment, we conducted an ablation study regarding the architectural modifications.
> As a result, for instance, we confirmed the effectiveness of employing the Pointer mechanism (l.445-457) by observing $1.7$ points improvement in keyframe selection score($\texttt{AKM}_{\texttt{ex}}$).
> Also, through the modification of the loss function (l.459-467), we observed an improvement of $0.3$ points in the caption score (BLEURT).
> We will include the details of this ablation study in the camera-ready version.
>
>
>
> [1] Khushboo Khurana and Umesh Deshpande. 2023. Two stream multi-layer convolutional network for keyframe-based video summarization. Multimedia Tools and Applications, pages 1–42.
>
> [2] Abhay Zala, Jaemin Cho, Satwik Kottur, Xilun Chen, Barlas Oguz, Yashar Mehdad, and Mohit Bansal. 2023. Hierarchical Video-Moment Retrieval and Step-Captioning. In Proceedings of the IEEE/CVF Conference on Computer Vision and Pattern Recognition
> (CVPR), pages 23056–2306.
>
> [3] Paul Hongsuck Seo, Arsha Nagrani, Anurag Arnab, and Cordelia Schmid. 2022. End-to-end Generative Pretraining for Multimodal Video Captioning. In Proceedings of the IEEE/CVF Conference on Computer Vision and Pattern Recognition (CVPR), pages 17959–17968.
>
> [4] Elyas Rashno and Farhana Zulkernine. 2023. Efficient Video Captioning with Frame Similarity-Based Filtering. In Database and Expert Systems Applications, pages 98–112.

---

### Official Review · Reviewer_ZpMr · 2023-08-06

**Typos Grammar Style And Presentation Improvements:** No typos that I noticed.
**Soundness:** 4

**Excitement:**

4: Strong: This paper deepens the understanding of some phenomenon or lowers the barriers to an existing research direction.

**Missing References:**

No missing references.

**Paper Topic And Main Contributions:**

This paper proposes a new video summarization task consists of jointly generating keyframes and appropriate captions so as to create a good summary consumable by a human being. The proposed approach cleverly exploits an existing dataset by adding additional attributes to the keyframes. The proposed baselines are sound and insightful. The proposed new task and the dataset are convincing, and set up a rewarding avenue for future research.

**Questions For The Authors:**

1. Nice work. Have you thought about how the new task you propose might vary across various content genres? For example, a sitcom often has little visual variation but has high textual variation while a sportscast might have very high visual variation. Just a question, not a requirement.

**Reasons To Accept:**

1. Thorough literature review including papers from pre-deep learning days.
2. Convincing new task developed through clever exploitation of existing dataset.
3. Solid and insightful baseline methods.


**Reasons To Reject:**

1. No significant weaknesses to report.

**Reproducibility:**

5: Could easily reproduce the results.

**Reviewer Confidence:**

5: Positive that my evaluation is correct. I read the paper very carefully and I am very familiar with related work.

---

> ### Author Rebuttal · Authors · 2023-08-29
>
> Thank you for your positive feedback.
>
> ## Question about the influence of the video genre
> Thank you for your insightful question.
> While we have not conducted a quantitative analysis, there were noticeable qualitative differences in the task's difficulty depending on the video genre.
> For instance, as you pointed out, in genres like sitcoms, the model frequently generated nearly identical captions repeatedly.
> We think this difficulty arises from the limited visual variations within the video, making it challenging for the model to interpret the contextual situation accurately (l.1037-1039).
> Conversely, we observed a tendency for the model to generate appropriate captions in genres like sportscasts.
> We expect that in genres characterized by limited visual variation, such as sitcoms, there is a potential for generating more appropriate captions using the audio information embedded in the video. This is our next direction for enhancing our video summarization system, but it exceeds the focus of the current paper.

---

### Meta-Review · Area_Chair_ztwG · 2023-09-21

**Recommendation:** 4

**Metareview:**

The paper proposes a framework for selecting relevant (key-)frames and corresponding captions, generated from videos that will constitute a summary of the video. The authors formulate the task as two main challenges: (1) the method of selecting the frames from video segments (solved using iterative and simultaneous approaches), and (2) evaluation of the task (aligned keyframe matching score for frames and METEOR for caption). The authors also contribute a convincing new dataset by adding additional attributes to the keyframes of existing data. The proposed baselines are sound and insightful, setting up a rewarding avenue for future research. The paper is overall well written, and authors commit to updating the paper in response to a meaningful discussion with reviewers.

---

### Decision · Program_Chairs · 2023-10-07

**Decision:**

Accept-Main

**Comment:**

The paper proposes a framework for selecting relevant (key-)frames and corresponding captions, generated from videos that will constitute a summary of the video. The authors formulate the task as two main challenges: (1) the method of selecting the frames from video segments (solved using iterative and simultaneous approaches), and (2) evaluation of the task (aligned keyframe matching score for frames and METEOR for caption). The authors also contribute a convincing new dataset by adding additional attributes to the keyframes of existing data. The proposed baselines are sound and insightful, setting up a rewarding avenue for future research. The paper is overall well written, and authors commit to updating the paper in response to a meaningful discussion with reviewers.